# The Effects of Oncological Treatment on Redox Balance in Patients with Uveal Melanoma

**DOI:** 10.3390/diagnostics13111907

**Published:** 2023-05-29

**Authors:** Mihai Adrian Păsărică, Paul Filip Curcă, Marian Burcea, Speranța Schmitzer, Christiana Diana Maria Dragosloveanu, Alexandru Călin Grigorescu

**Affiliations:** 1Clinical Department of Ophthalmology, “Carol Davila” University of Medicine and Pharmacy, 050474 Bucharest, Romania; m.pasarica@yahoo.com (M.A.P.); mnburcea@gmail.com (M.B.); speranta.sch@gmail.com (S.S.); christianacelea@gmail.com (C.D.M.D.); 2Department of Ophthalmology, Clinical Hospital for Ophthalmological Emergencies, 010464 Bucharest, Romania; 3Department of Oncology, Institute of Oncology Prof. Dr. Alexandru Trestioreanu, 022328 Bucharest, Romania; alexgrigorescu2004@yahoo.com; 4Department of Oncology, Clinical Hospital of Nephrology Dr. Carol Davila, 010731 Bucharest, Romania

**Keywords:** uveal melanoma, redox balance, free radicals, oxidative stress

## Abstract

(1) Background: Uveal malignant melanoma is the most common adult eye cancer and presents metabolic reprogramming that affects the tumoral microenvironment by altering the redox balance and producing oncometabolites. (2) Methods: The study prospectively evaluated patients undergoing enucleation surgery or stereotactic radiotherapy for uveal melanoma by following systemic oxidative-stress redox markers serum lipid peroxides, total albumin groups and total antioxidant levels (3) Results: Serum antioxidants and lipid peroxides were elevated from pre-treatment to longer-term follow-up. Antioxidants inversely correlated to lipid peroxides: higher in stereotactic radiosurgery patients pre/6/12/18 months post-treatment (*p* = 0.001–0.049) versus higher lipid peroxides in enucleation surgery patients pre/after/6 months post-treatment (*p* = 0.004–0.010). An increased variance in serum antioxidants was observed for enucleation surgery patients (*p* < 0.001), however enucleation did not increase mean serum antioxidants or albumin thiols; only lipid peroxides were increased post-enucleation (*p* < 0.001) and at 6-month follow-up (*p* = 0.029). Mean albumin thiols were increased for 18- and 24-month follow-ups (*p* = 0.017–0.022). Males who had enucleation surgery presented higher variance in serum determinations and overall higher lipid peroxides values pre/post-treatment and at the 18-month follow-up. (4) Conclusions: Initial oxidative stress-inducing events of surgical enucleation or stereotactic radiotherapy for uveal melanoma are followed by a longer-term inflammatory cascade gradually subsiding at later follow-ups.

## 1. Introduction

Uveal malignant melanoma (UMM) is the most common primary intraocular cancer in adults [1], with an incidence 10 to 20 times lower than cutaneous melanoma [1] and a peak incidence around the age of 50–60 [1]. Ocular treatment of uveal melanoma can be “conservative” if the treatment is aimed at conserving useful vision [2] or “radical” if consisting of surgical enucleation [2]. Radiotherapy is a common conservative treatment [3] which uses stereotactic radiosurgery techniques such as gamma-knife and cyber-knife devices, high-precision proton beam, or plaque brachytherapy.

### 1.1. Epigenetic Differences of Uveal Malignant Melanoma Compared to Cutaneous Melanoma

UMM is genetically distinct from cutaneous melanoma (CM): while CM cells carry mutation of *BRAF*, *NRAS* or *KIT* genes, UMM cells carry activating mutations in the G protein-coupled receptor (GPCR) pathway alpha subunits *GNAQ* or *GNA11* [4,5] and inactivating somatic mutations in gene encoding for *BRCA-1-associated protein 1 (BAP1)* [4,5]. Due to *BAP1* inactivation mutations, UMM is predisposed to metastasis [4,5], with 84% of *BAP-1* mutated patients developing liver (89%), lung (29%) and bone (17%) secondary determinations [4], which can appear in up to 40–50% of UMM patients despite early diagnosis and treatment [4]. Integrative analysis studies of UMM gene expression have reported higher mutation frequency of *GNA11* in the 3-monosomy subtype [6] and *BAP1* alterations in as high as 85% of 3-monosomy UMM [7] with novel sequence assembly methods [7].

### 1.2. Epigenetic Changes in Turn Produce Metabolic Reprogramming of UMM Cells

The Gq/11 subfamily of heterotrimeric G protein α-subunits was found by Onken et al. to be the principal driver of metabolic reprogramming [8] by encoding genes for nearly all enzymes involved in glycolysis, the tricarboxylic acid cycle (TCA, Krebs cycle) and oxidative phosphorylation in patient-derived tumor cells [8]. As such, Gq/11 driven metabolic reprogramming increases glycolysis and glucose uptake [8], enhances mitochondrial respiration and is required to chronically sustain metabolic reprograming of the malignant uveal melanoma cells [8]. UMM has high glycolytic activity under 18-F-Fluorodeoxyglucose positron emission/computed tomography (PET/CT) [4]. Monosomy 3 UMM has a lower gene expression profile related to glycogen synthesis and lower amounts of glycogen in tumor tissues [4]; however, it exhibits a systemic effect through insulin-resistance generation. This assures higher levels of available fasting plasma glucose levels [4] and lower adiponectin levels [4], with metastatic and high-risk monosomy 3 UMM patients having much lower adiponectin levels [4].

### 1.3. Oxidative Stress in Uveal Malignant Melanoma

Oxidative stress is defined as an imbalance between the production of reactive oxygen species (ROS) (reactive metabolites and free radicals) and the eliminative protective processes, the antioxidants [9]. Sustained oxidative stress damages cell structure and functions and is oncogenic [4,9]. Inside the tumoral microenvironment (TME) uveal melanoma cells try to maintain a favorable redox balance for their survival and create an environment where the most aggressive malignant populations outcompete other variants and native tissue cells to propagate and disseminate [4,10]: Onken et al. found that when selectively inhibiting the oncogenic Gq/11 signaling which is responsible for the elevated reprogrammed metabolism, the tumoral cells adapt by activating genes involved in facilitating nutrient scavenging and maintenance of redox homeostasis [8] that could purportedly promote UMM cell survival [8].

Succinate dehydrogenase A *(SDHA)* is a component of Complex II of the electron transport chain (ETC) [11] and an essential link between the TCA cycle and OXPHOS. *SDHA* oxidizes succinate, liberating two electrons for shuttling to the C Cytochrome (CYC1), facilitating OXPHOS [11]. Regarding mitochondrial metabolism, UMM cells display the highest median oxidative phosphorylation (OXPHOS) gene expression level among other analyzed cancers [4] by highly upregulating the *macroH2A1* histone variant [4]. This is contrary to cutaneous melanoma, in which *macroH2A1* proteins actually suppress progression as reported by Kapoor et al. [12] and Giallongo et al. [13], and is particular to uveal melanoma. Monosomy 3 UMM has even more active mitochondria and a higher mitochondrial reserve capacity [4]. UMM cell behavior is relatively distinct from cutaneous melanoma and does not follow a classic Warburg effect loop [14] where high glycolysis leads to excess lactate that alters the redox balance further, acting as an oncogenic factor [5,10]. Longhitano et al. [5] observed that lactate supplementation impairs tumor growth acting via MCT1 rather than modulating the HCAR1 cascade; however, lactate supplementation also boosts transporters and crucially increases OXPHOS activity. Mutations in *SDH* genes such as *SDHA* and others metabolic genes [11] produce “oncometabolites” which promote tumorigenesis [11] and alter the oxidative stress balance. Blasi et al. found that uveal melanocytes are susceptible to peroxidative stress and deleterious effects of prooxidants [14] due to an imbalance in the superoxide-dismutase (SOD)/catalase ratio (alteration of the scavenger system) [14]. The study noted that the higher the proliferation rate of less differentiated cells, the lower the total antioxidant protection system present [14], and that higher membrane PUFA percentage and vitamin E levels partially compensate for this vulnerability [14]. As such, free-radical mediated damage could be both an oncogenic promoter [14] and a potential metabolic vulnerability of UMM cells.

Metabolic reprogramming in uveal malignant melanoma is distinct and more concentrated towards mitochondrial oxidative oncometabolism than just the Warburg effect. This discrepancy in the oxidative and redox behavior of UMM warrants further study of UMM and redox interaction via systemic redox parameter measurement or, if possible, via microanalysis of the tumoral environment. Finally, further research on the therapeutic potential of metabolic inhibitors [4,15] could offer suppression of *BAP1*-mutant UMM subtypes [4] and reduce malignant cell survival via OXPHOS inhibitors [4].

## 2. Materials and Methods

### 2.1. Aim of the Study

The prospective study aims to steudy the effect of uveal malignant melanoma treatment on systemic redox and oxidative-stress parameters. Although several studies such as Han [4], Longhitano [5], Onken [8] and Chattopadhyay [11] discussed aspects of metabolic reprogramming in UMM, the tumoral microenvironment and its distinct difference from cutaneous melanoma, to our knowledge, there is a lack of data regarding the systemic effects on oxidative and redox balance in UMM. We aimed to monitor the above-mentioned redox balance via measurement in serum levels of lipid peroxidation markers, total albumin groups and total antioxidant levels for patients undergoing uveal melanoma treatment via stereotactic radiotherapy or enucleation. The patients were followed for a period of 2 years, with serum measurements taken before and after treatment and at subsequent 6-month intervals. The therapeutical decision of radiotherapy or enucleation was performed in each case by the oncological commission based on patient parameters, independent of the study. Accordingly, 21 patients received stereotactic radiotherapy while 18 patients received enucleation treatment.

### 2.2. Study Inclusion and Exclusion Criteria

The inclusion criteria for the study were: age between 18 and 70 years, confirmed clinical and paraclinical diagnosis of malignant uveal melanoma, Eastern Cooperative Oncology Group (ECOG) scores between 0 and 2. All patients were enrolled after providing written informed consent for study participation, which could be withdrawn at any point during the study should the patient choose to. The following patients were excluded from enrollment into the study population: age outside 18–70 interval, altered clinical status with extensive metastatic disease throughout the body, other complex oncological treatment, patient behest to not participate or withdrawal of consent at any point during the study.

### 2.3. Laboratory Determinations Methodology

After providing written consent, venous punction blood samples were taken from the patients. The samples were centrifugated and the resultant serum isolate was used for laboratory determinations. The following markers were determined:serum lipid peroxides in micromoles (μmol)/100 mililiters (mL) by measuring the reaction of serum malondialdehyde (MDA) [16,17] as a final product of lipid hydroperoxide degradation [16]. We evaluated by reaction with 2-tiobarbituric acid (TBA) as determined through the Carbonneau method [18]. The chemicals used were: solution of TBA (Sigma) acid 0.7% dissolved in acetic acid 50%; trichloride acetic acid 20% (Sigma); Buffer solution acetic acid—sodium acetate 50 mmols, pH value of 7.total serum thiol-albumin groups micromoles (μmol)/100 mililiters (mL) measurement using the Albini method [19] and Ellman reagent substrate (5,5′-dithiobis-(2-nitrobenzoic acid, DTNB) [20] measurement of oxidative protein degradation [20]. The Ellman reactive was prepared by dissolving, at warmth, 4% 5,5′-dithiobis-(2-nitrobenzoic acid (DTNB) in 100 mL of buffer phosphate solution with pH value of 8.0. After reacting plasma serum isolate with DTNB, the resultant compound was measured using spectrophotometry.total level of antioxidants in micrograms (μg)/liter (L), inferred via measurement of the ferric reducing ability of serum (FRAS) [21,22]. We prepared an acetate buffer solution 300 millimole (mM) pH 3.6 with 2,4,6-tri(2-pyridyl)-1,3,5-triazine (TPTZ) 10 mM and ferric chloride 20 mM in distilled and pure water obtained using the Millipore system (Milli-Q-Biocel, MilliporeSigma, subsidiary of Merck, Munich, Germany). 1 mM ferrous sulphate was used to prepare standard solutions for the etalon curve. The ferric reducing ability of plasma (FRAP) reactive was prepared from 200 mL of buffer acetate solution (pH 3.6), TPTZ solution, FeCl_3_ 6H_2_O and distilled and purified water. For 30 μL of serum plasma probe 900 μL of FRAP reagent and 90 μL of water were added. The resulting compound was determined using spectrophotometry.

The compounds used were of analytical purity (stated 98% purity) and produced by Sigma and Merck (Merck, Burlington, MA, USA) (Sigma-Aldrich, subsidiary of Merck, Munich, Germany). Genetically pure water was used, obtained using a Millipore (Milli-Q-Biocel) system (MilliporeSigma, subsidiary of Merck, Munich, Germany), and the spectrophotometry determinations were performed on a Specord 210 (Analytik Jena GmbH, Jena, Germany) device.

### 2.4. Statistics

#### 2.4.1. Reference Value Comparison

Reference laboratory values provided were set at 0–4 μmol/100 mL of serum for lipid peroxides, 350–450 μmol/100 mL for thiol-albumin groups and 0.9–1.6 μg/L for total antioxidant levels. The statistical software Minitab^®^ 20 (Minitab Ltd., Coventry, UK) and SPSS version 29 (the current, subscription-based version) (International Business Machines Corporation, IBM, Endicott, NY, USA) were used for the statistical analysis of the study data. To evaluate the significance of the determinations against reference values, we used the one sample *t* test and the sign test for a median to compare the groups against a specified median, in this case the maximal admissible reference value (Table 1).

#### 2.4.2. Pre/Post-Treatment and 6/12/18/24-Month Follow-Up Comparison

Serum determinations of antioxidants (μg/L), lipid peroxides (μmol/100 mL) and albumin thiols (μmol/100 mL) were taken immediately before commencing treatment (average of 7 individual determinations), post-treatment and at 6, 12, 18, and 24 months post-treatment. For each individual column series of data (time interval—type of serum determination; for example, one column series was antioxidants at 6 months post-treatment), descriptive statistics data were analyzed and normality testing was completed. Data distribution was found to be normal for all column series using the Anderson–Darling (AD) normality test and, for antioxidants post-treatment and albumin thiols at 6 months post-treatment, the Kolmogorov–Smirnov (KS) and Ryan–Joiner/Shapiro–Wilk (RJ) tests (Table 1). For comparison purposes, the paired samples *t* test was used to compare pre-treatment to post-treatment results.

#### 2.4.3. Enucleation Surgery vs. Stereotactic Radiotherapy Comparison 

For this comparison, the data were divided into individual enucleation surgery (ES) at pre/post-treatment/6/12/18/24-month groups and stereotactic radiotherapy (SR) groups at the same time intervals. Comparison of ES and SR groups utilized the independent samples *t* test (SPSS 29) or two samples *t* test (Minitab 20). For variance comparison, analysis of variance (ANOVA) testing was used.

Analysis of the treatment dynamics of ES and SR was performed for the entire study population at 6 months and the complete 2-year follow-up population using Bayesian one-way repeated measure ANOVA weighted by treatment type (enucleation surgery ES and stereotactic radiotherapy SR).

Correlations between the pre-treatment and subsequent determinations were evaluated using Pearson correlations, with an ideal correlation represented by a PPC equal to +1 or −1 with statistical significance (*p* < 0.05).

#### 2.4.4. Study Limitations

We thus studied in dynamic the effect of the oncological treatment on the systemic redox balance of UMM patients for up to 2 years by focusing on serum measurements performed in our laboratory department. Thus, one of our study’s limitations is concentrating on laboratory determinations versus following associated clinical data such as visual acuity or quality-of-vision related questionnaires. The prospective design of the study connected with another study limitation: the limited patient cohort of 39 patients, which could be improved by supplementary multi-department follow-up studies. Finally, the study compared the serum determinations with reference values and not with a second, control group without UMM. We acknowledge these study limitations.

#### 2.4.5. Local Ethics Committee Approval

The study and publishing of the study results were approved by the Local Ethics Committee for Scientific Research of the Oncological Institute Prof. Dr. Alexandru Trestioreanu Bucharest (3634, submitted 22 March 2023 and approved 11 May 2023). Following the analysis of the study conceptualization, development, execution, and submitted study and patient documents which contain requests to use data from clinical presentation and history, anamnesis, previous hospital presentations, diagnostic data, laboratory data, data from clinical, imagistic, and other investigations, postoperative and other follow-up data, the Ethics Committee approved of the aforementioned publication of the study.

## 3. Results

### 3.1. Study Population and Follow-Up

The study population was 39 patients, including 17 females (43.58%) and 21 males (56.41%), who underwent treatment for uveal melanoma either by stereotactic radiosurgery (21 patients, 11 females (52.38%) and 10 males (47.69%)) or by surgical enucleation (18 patients, 6 females (33.33%) and 12 males (66.66%)). In the study’s duration of 2 years, follow-up intervals were set 6, 12, 18 and 24 months after treatment. Respectively, 39 patients (100%) attended follow-up at 6 months, 22 patients (56.41%) at 12 months (11 females and 11 males), 11 patients (28.20%) at 18 months (5 females 45.45% and 6 males 54.54%) and 6 patients (15.38%) at 24 months (4 females 66.66% and 2 males 33.33%). The study tracked laboratory determinations throughout the follow-up intervals. The mean time of loss of follow-up was on average 11.85 months for all patients, 13.06 months for females only and 10.91 months for males. Loss of follow-up was attributed to poor health, decease, difficulties accessing the medical center and individual desire to not participate in further follow-up.

### 3.2. Mean Serum Determinations Results in Dynamic vs. Baseline Reference

Serum antioxidants were found to be statistically increased before and after treatment until the last 24-month follow-up (*p* between <0.001 and 0.049). Serum lipid peroxides increased both before and after treatment until the 18-month follow-up (<0.001–0.001). Most importantly, and as summarized in Figure 1, serum antioxidants spiked upwards immediately after treatment and gradually decreased at subsequent follow-ups, returning no statistical difference at 24-month follow-ups. Serum lipid peroxides trended downwards from the initial plateau until no statistical difference versus the reference could be noted (Figure 2). Albumin thiols were not statistically elevated for any serum determination versus the maximal admissible reference value (Table 1; represented in Figure 3). When separated by procedure, albumin thiols in stereotactic radiosurgery patients appeared to trend downwards from the initial plateau, while albumin thiols in enucleation surgery patients slightly varied upwards (Figure 3); this was explored in detail and found to be not statistically significant; albumin thiols were comfortably in the reference interval for the 24-month follow-up (79.03% of maximal admissible reference, *p* = 0.039).

### 3.3. Comparing Serum Determinations of Enucleation Surgery (ES) vs. Stereotactic Radiotherapy (SR) Patients

#### 3.3.1. Antioxidants

Both starting, pre-treatment values of serum antioxidants (1.703 μg/L mean for ES and 2.508 μg/L mean for SR) were above reference values (Table 1). Between the pre-treatment enucleation surgery (ES) and stereotactic radiotherapy (SR) groups, there was a statistically significant difference (*p* < 0.001), with 0.805 μg/L higher values in the patient group proposed for SR versus ES. We observed a 92.041% increase in the mean serum antioxidants levels to 3.271 μg/L in the ES group; however, this was offset in significance by a higher increase in the variance between individual measurements (*p* < 0.001, Table 2). Due to this increased variance, statistical testing did not produce a clear difference between pre/post-treatment measurements when comparing ES to SR (*p* = 0.602–0.633). Thus, we note an increase in the variance of post-enucleation surgery results (*p* < 0.001) and observe that both treatments were just as effective based on the criteria of systemic serum antioxidants levels. Overall, mean serum antioxidant values were found to be statistically increased before and after treatment until the last 24-month follow-up (*p* between <0.001 and 0.049, Table 1). Analysis using Pearson correlations produced the least conclusive correlation between pre/post-treatment antioxidant levels (0.101 PPC), while the other PPC values for antioxidants were closer to an ideal correlation (−1 or 1 PPC) as the length of follow-up increased.

We explored in detail the statistical impact of the treatment type (Table 2) or patient gender (Table 3). Stereotactic radiosurgery patients presented statistically higher serum antioxidants mean values pre-treatment and at 6-, 12-, and 18-month follow-ups; for post-treatment levels, the enucleation surgery group had higher values, but as previously discussed, this difference was not statistically significant (*p* = 0.633 < 0.05; Table 2) due to variance increases. Patient gender did not influence antioxidant values at each follow-up interval (Table 2).

#### 3.3.2. Lipid Peroxides

For lipid peroxides, the pre-treatment and post-treatment/6/12 serum determinations were above the maximal reference value (Table 1). The starter pre-treatment measurements presented differences when compared by the proposed treatment option: +1.305 μmol/100 mL higher in patients proposed for ES, *p* = 0.004 (Table 2). The higher lipid peroxide serum values remained so in ES patients for the post-treatment and 6-month follow-ups (*p* between 0.004–0.010, Table 2). Patient gender also influenced serum determinations, with higher measurements in males pre/post-treatment (*p* = 0.029/0.134) and at the 18-month follow-up (*p* = 0.031). The *p* value for the 6-month follow-up, 0.054, was close to the 95% confidence interval (CI) and would have been valid with a larger 90% CI. Thus, serum lipid peroxide values were sensitively higher in enucleation surgery and male patients up to and including the 6-months post-operative follow-up. Interestingly, in the previous analysis for antioxidant levels, it was stereotactic radiotherapy which had higher post-treatment levels, apart from the measurements snapshot performed immediately post-treatment (Table 2). This could imply an approximate inverse correlation relation between antioxidants and lipid peroxides when filtered by treatment type.

Lipid peroxide levels were higher at 6- and 12-month follow-ups compared to pre-treatment levels (*p* = 0.001 and 0.029, respectively, Table 4). For the pre/post-treatment comparison, no statistical difference was found (*p* = 0.120). The variance of the groups did not differ (Table 4). Despite these findings, the Pearson correlations showed a strong correlation between pre-treatment serum lipid peroxides μmol/100 mL and post-treatment/6/12/18/24-months values with PPC values above 0.9 which approached an ideal correlation of PPC = 1; these correlations were all statistically significant (minimum *p* < 0.001, maximum *p* = 0.002). Overall, when comparing the pre-treatment study population to follow-ups, serum lipid peroxides were modified in the 6 to 12 month post-treatment window (Table 4) and were influenced by the treatment type and patient gender.

#### 3.3.3. Albumin Thiols

For albumin thiols a statistical difference was found in the 18/24 months post-treatment groups when grouping by procedure (Table 2). No statistical difference was found in pre/post-treatment group variances (Table 2). Comparing pre-treatment to follow-up measurements did not produce statistical differences (Table 4). Even when comparing pre-treatment serum albumin thiols with 24 months follow-up results, no difference was noted (*p* = 0.524). We observe that albumin thiols were the weakest of the three serum parameters for following the treatment dynamic (versus antioxidants and lipid peroxides).

#### 3.3.4. Treatment Dynamics for the First 6 Months

For patients with 6 months of follow-up (entire study population, 39), analysis of the data weighted by the treatment type (enucleation surgery ES and stereotactic radiotherapy SR) revealed a statistically significant variance difference (Bayes factor 0.025, *p* < 0.001) with peak antioxidants variance post-treatment (95% credible interval lower limit of 2.309 μg/L and upper bound of 3.27 μg/L, mean 2.79 μg/L versus pre-treatment Bayesian mean 2.26 μg/L and 6 months post-treatment Bayesian mean of 2.33 μg/L). The fluctuation in variance was also present when weighing the results by patient gender due to a mean increase in post-treatment antioxidant serum levels (Bayes factor 0.078, *p* < 0.001; Bayesian mean 3.09 μg/L versus 2.14 μg/L pre-treatment and 2.18 μg/L post-treatment). For lipid peroxides and albumin thiols, neither treatment type nor patient gender affected the variance of the dynamic determinations. Repeat measurements ANOVA by procedure were lipid peroxides Bayes factor 508.201, *p* = 0.378 > 0.05, albumin thiols Bayes factor 0.345, *p* = 0.273 > 0.05; repeat measurements ANOVA by patient gender were Bayes factor 7.193, *p* = 0.190 > 0.05 for lipid peroxides, Bayes factor 0.197, *p* = 0.327 > 0.05 for albumin thiols.

#### 3.3.5. Treatment Dynamics for the Complete 2-Year Study Period

For the patients with complete 2-year follow-ups (6), analysis of the data revealed no statistically significant differences between post-treatment serum measurements for the study population when the data were weighted by treatment type (antioxidants *p* = 0.214 > 0.05; lipid peroxides *p* = 0.573 > 0.05; albumin thiols *p* = 0.234 > 0.05) or by patient gender (antioxidants *p* = 0.121 > 0.05; lipid peroxides *p* = 0.757 > 0.05; albumin thiols *p* = 0.373 > 0.05). Our findings here could be limited by the small 2-year follow-up population (6) in comparison to the 6-month follow-up study population (39).

## 4. Discussion

Our study followed the impact of two-year treatment dynamics on systemic oxidative stress metabolites and was limited in patient population scope or control groups due to the prospective design. Analysis of the values recorded in the study indicates important increases of stress on the antioxidant metabolic systems due to the presence and treatment of uveal malignant melanoma. Circulating serum antioxidants were statistically elevated for all but the last follow-up at 24 months post-treatment determinations versus the reference; serum lipid peroxides were increased both before and after treatment until the 18-month follow-up. When compared to baseline pre-surgery values, an increased variance in serum antioxidants after tumor-resection surgery (surgical enucleation) was confirmed. Serum lipid peroxides were increased post-treatment and at 6 months post-treatment and hence influenced by the application of the treatment. This effect could be implied to continue via elevated antioxidant and lipid peroxides levels up to 12 to 18 months post-treatment, long after the initial therapy. Thus, oxidative stress is first immediately influenced by surgical trauma or apoptotic effect of stereotactic radiotherapy. However, the presence of UMM and subsequent treatment procedures trigger a long-term cascading reaction producing circulating free radicals which is ongoing long after tumor removal has occurred and would correlate with observed elevated results in the follow-up groups.

Albumin thiols behaved differently than antioxidants and lipid peroxides in our study. When including patients that benefited from both enucleation surgery and stereotactic radiotherapy, no significant increases could be detected for all follow-up intervals. When comparing at certain follow-up interval patients that received ES to those that benefited from SR, the shorter-term follow-up was not affected (Table 2); however the measurements did exhibit an observable increase in mean values and variance for ES patients from our cohort at 18- to 24-month follow-ups. Thiol groups are involved in the detoxification of radical oxidative species (ROS) [23]. Hanikoglu et al. observed a similar increase in prostate cancer patients with elevated thiol and native thiol levels at 6 months after radical prostatectomy [23] and indicated that due to native thiol loss and deteriorating capacity to resist oxidant stress the thiol/disulphide homeostasis begins to shift towards replenishment of thiol levels following tumoral removal surgery [23], thus correlating with postoperatively elevated levels.

Several studies have approached the in vivo therapeutical effects of antioxidative and pro-oxidant therapy [24]. Intracellular ROS and oxidative stress are pro-oncogenic [24] and treatment of the oxidative imbalance could reduce tumorigenic properties [25]. Furthermore, ROS were noted by Guijarro [25] as signaling molecules in tumorigenesis. However, ROS are also destructive on tumoral cells in higher concentration; Afzal et al. noted that paclitaxel, an established mitotic inhibitory drug [26], promotes ROS generation by enhancing the activity of NADPH oxidase NOX [25] with accumulation of ROS products outside the cells which provokes lethal damage to bystander cancer cells that have not been exposed to paclitaxel [26,27]. Finally, increases in total antioxidant levels (excess of reactive oxygen species ROS) show a lasting inability and overcoming of capture and counteraction systems [16]. Gradual increase after therapy in these values, when taken together with the clinical picture, could even be a reason for suspicion of neoplastic disease progression. Overall, antioxidant parameters monitoring could provide data with clinical impact and more research is needed on the topic.

Liu-Smith et al. [28] studied the gender differences in uveal malignant melanoma on immune response and redox regulation. The PDIA2 protein, an endoplasmic reticulum-located glycoprotein [28], was noted in their study as both a possible estrogen regulator enzyme [28] and an immune-functioning protein that can directly bind to the human major histocompatibility complex class 1 antigens (HLA-A, B and C) [28]. This protein was downregulated in male tumors in comparison to female tumors [28], the implication being that estrogen helps to deal with oxidative stress [28]. Males with UMM also overexpressed several protein coding genes [28]: *Immunoglobulin Kappa Locus IGK*, *Immunoglobulin Lambda Like Polypeptide 5 (IGLL5)*, *cluster of differentiation CD79a* (known as B-cell antigen receptor complex-associated protein alpha chain) and *joining chain of multimeric IgA and IgM (JCHAIN)* [28].

In our study, antioxidants fluctuated in variance due to a post-operative peak which affected predominantly male + enucleation-surgery patients (repeated measures ANOVA, *p* < 0.001). Males had overall higher mean values of serum lipid peroxides at pre/post-treatment (*p* = 0.029 and 0.148) and at 18 months post-treatment (*p* = 0.031); at 6 months, the increase did not fit within the 95% CI (*p* = 0.054). Albumin thiols were not affected by patient gender. Overall, patient gender influenced post-operative variance of antioxidant serum levels and lipid peroxides pre/post/18 months post-treatment.

In 1978 Zimmerman [29], an eminent American pathologist, published an influential article that suggested that enucleation (and by implication, other surgical resections) accelerated metastatic death by physically disseminating tumor cells from the eye into the general circulation [29] based on observed mortality rate peaks in the second post-operative year [29]. These concerns lead to adoption of preventative measures, such as pre-enucleation radiotherapy [30]. To resolve controversies about Zimmerman’s [29] and Manschot’s hypotheses [31], forty centers in North America undertook a large, randomized, multi-center, Collaborative Ocular Melanoma Study (COMS) [32]. One of the studies of COMS investigated the impact of pre-enucleation external beam radiotherapy on survival in 1003 patients with large uveal melanoma [32], finding no survival advantage attributable to pre-enucleation radiotherapy [32]. Thus, the most plausible peak for observed high peak in metastatic death in the second post-operative year could be related to late patient presentation, with large tumor diameters such as 13 mm [30]. Straatsma et al. compared 43 untreated patients with historical controls [33] and reported a trend towards higher mortality in patients where treatment was deferred and not immediately applied; however, this could have been caused by selection bias towards older patients that had deferred treatment [33]. Gragoudas et al. have shown local tumor recurrence after conservative treatment to associate with higher mortality [34] and have suggested increased aggressiveness of recurrences with a shortened lifespan [34]; however, other authors [30] suggested the results could show selection of more aggressive uveal melanoma types.

The landmark COMS Study [32] previously reported that pre-enucleation radiotherapy did not improve clinical outcomes or mortality [32] and that plaque iodine-125 brachytherapy was an equally effective treatment option compared to enucleation surgery [35] for patients which respected the COMS study inclusion criteria (uniocular disease, age 21 or older, no coexisting disease that threatened survival for 5 years or longer, free of metastatic disease or other cancers and at least 20/200 vision in the healthy, other eye [35]). Radiotherapy methods for uveal melanoma advanced considerably towards the present option of high-dose and high-precision stereotactic radiotherapy [3] or proton beam therapy [3]. The COMS study followed predominantly tumor specific parameters such as apical height and longest basal diameter [31,35]; however, further research [5,6,7,8,9,10,11,12,13] into uveal melanoma has highlighted the importance of oncogenetic and metabolic reprogramming, with Honavar et al. arguing that UMM prognosis and treatment is increasingly governed by gene expression profile analysis [36] and detection of high-risk for metastasis disease such as monosomy-3 tumors [36]. In this new context of specific metabolic alterations, redox alterations and the tumoral microenvironment could be further research directions with potential future clinical implications.

Our study signals that redox serum parameters can be influenced by the presence of the uveal melanoma neoplastic process and subsequent treatment. To our knowledge, there is a lack of data on redox parameters in uveal melanoma versus other types of cancers where the significance of oxidative-stress parameters has been explored and compared against benign or inflammatory conditions. Kaya et al. reported clinical significance of oxidative parameters such as elevated MDA activity [37] (corresponding with elevated lipid peroxides in our study) in prostate cancer and benign prostatic hyperplasia (BHP) [37] as opposed to asymptomatic inflammatory prostatitis patients. Aydin et al. reported alteration in the lipid peroxidation index and concomitant antioxidant changes in prostate cancer [38] and recommended further research to determine if oxidative stress-related parameters could be used as differential diagnostic and prognostic tools in prostate cancer and BHP [38]. Higher serum or plasma lipid peroxidation markers have also been observed by Mazzuferi et al. [39] in breast cancer patients. The successful use of an oxidative stress score system (SOS) for assessing the prognosis of patients has been reported by Zhang et al. in operable breast cancer [40] and by Qian in early-stage lung adenocarcinoma [41]. In cutaneous melanoma, further research has been conducted for SOS impact [42] and for oxidative-stress prognostic genetic biomarkers [43]. Future research into integrating systemic oxidative stress parameters into a scoring system for uveal melanoma could, potentially, provide useful clinical prognostic or follow-up information.

## 5. Conclusions

Our study strives to address, to our knowledge, a lack of information related to systemic redox effects of uveal malignant melanoma and its treatment. The study observed inference on long-term oxidative stress parameters due to the presence and subsequent treatment of uveal malignant melanoma. Generally speaking, serum antioxidants and lipid peroxides were elevated pre-treatment and for a longer-term post-treatment duration. Specifically, levels of antioxidants were above the reference value from pre-treatment to 18 months post-treatment and lipid peroxides were above the reference value from pre-treatment to 12 months post-treatment. When the results were compared by treatment type, an inverse relationship between antioxidant and lipid peroxide levels was observed: higher antioxidants for stereotactic radiotherapy patients pre-treatment and at 6/12/18 months post-treatment versus higher lipid peroxides in enucleation surgery patients pre-treatment and post-treatment and at 6- and 12-month follow-ups. Albumin thiols were only elevated at 18- and 24-month follow-ups in the enucleation surgery group and could indicate replenishment of thiol groups following oxidative stress. A patient gender difference was noted in the study with males who had enucleation surgery having had higher variance in serum determinations and overall higher lipid peroxides values pre/post-treatment and at the 18-month follow-up.

While the oxidative stress markers studied differed at the various follow-up intervals, taken together, they form a picture of an initial oxidative stress-inducing event of surgical enucleation or applied stereotactic radiotherapy which is followed by a longer-term inflammation cascade that gradually subsides at later follow-up. The dynamic alteration of systemic serum oxidative stress markers at different moments in the treatment protocol warrants further research with regards to theoretical and possible clinical implications in uveal melanoma, since similar studies have been performed for prostate, breast, or lung cancers [38,39,40,41,42,43]. Integration of systemic oxidative stress parameters into a scoring system for uveal melanoma could be beneficial for assessing prognosis.

## Figures and Tables

**Figure 1 diagnostics-13-01907-f001:**
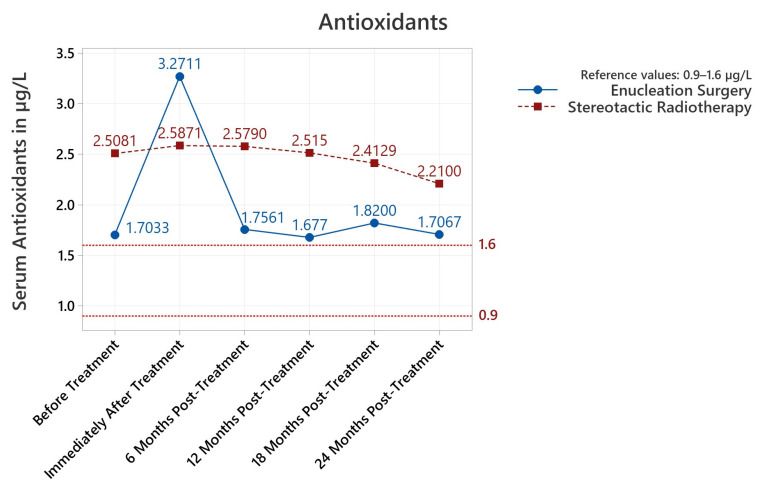
Summary of mean values from serum determinations of antioxidants (μg/L) for each time interval of the study. The reference interval is represented by dotted red lines (0.9–1.6 μg/L).

**Figure 2 diagnostics-13-01907-f002:**
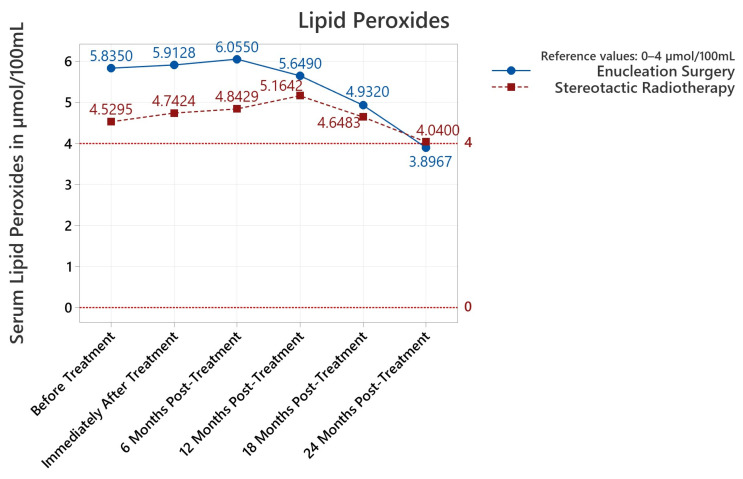
Summary of mean values from serum determinations of lipid peroxides (μmol/100 mL) for each time interval of the study. The reference interval is represented by dotted red lines (0–4 μmol/100 mL).

**Figure 3 diagnostics-13-01907-f003:**
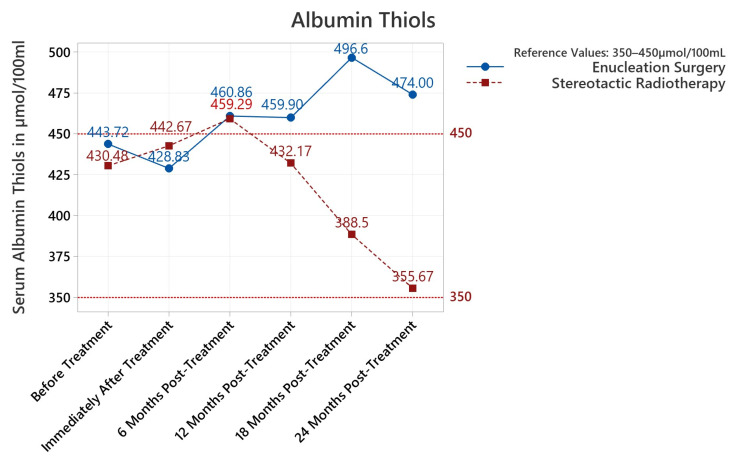
Summary of mean values from serum determinations of albumin thiols (μmol/100 mL) for each time interval of the study. The reference interval is represented by dotted red lines (350–450 μmol/100 mL).

**Table 1 diagnostics-13-01907-t001:** Serum results and treatment dynamic.

	Antioxidants Reference 0.9–1.6 μg/L					Lipid Peroxides Reference 0–4 μmol/100 mL					Albumin Thiols Reference 350–450 μmol/100 mL				
Treatment Dynamic	Mean	% inc *	*p*	Significance	Values over Ref	Mean	% inc *	*p*	Significance	Values over Ref	Mean	% inc *	*p*	Significance	Values over Ref
Pre-Treat	2.137	133.56%	<0.001	Yes	33/39	5.132	128.30%	<0.001	Yes	28/39	436.6	97.02%	0.279	No	17/39
Post-Treat	2.903	181.43%	0.049	Yes	35/39	5.283	132.07%	<0.001	Yes	33/39	436.3	96.95%	0.511	No	15/39
6 Months	2.199	137.45%	<0.001	Yes	32/39	5.402	135.05%	<0.001	Yes	33/39	460	102.22%	0.624	No	21/39
12 Months	2.134	137.37%	<0.001	Yes	18/22	5.385	134.62%	0.001	Yes	19/22	444.8	98.84%	0.734	No	9/22
18 Months	2.166	135.37%	0.002	Yes	10/11	4.777	119.42%	0.144	No	8/11	437.6	97.24%	0.633	No	5/11
24 Months	1.958	122.37%	0.082	No	5/6	3.968	99.2%	0.952	No	2/6	414.8	92.17%	0.290	No	2/6

* A percentage increase has been calculated using the maximum admissible reference value for each group of studied data (Antioxidants 1.6 μg/L, Lipid Peroxides 4 μmol/100 mL, Albumin Thiols 450 μmol/100 mL). Using the one sample *t* test, a comparison was made to the maximum admissible reference value (“*p*” and “Significant?” rows). A *p* value < 0.05 was considered statistically significant. The “Significance” column was completed Yes if the *p* value indicated statistical significance. Using the sign test for a median, the number of measurements above maximum admissible reference values was quantified. Abbreviations used: μmol/100 mL—micromoles/100 milliliters, μg/L—micrograms/liter, Ref—Reference values, % inc—Percentage increase versus maximal admissible reference values, *p*—*p* value of the one sample *t* test versus maximal admissible reference.

**Table 2 diagnostics-13-01907-t002:** Statistical study of the inference of treatment type on serum levels of antioxidants, lipid peroxides and albumin thiols.

Enucleation vs. Radiotherapy	Antioxidants μg/L			Lipid Peroxides μmol/100 mL			Albumin Thiols μmol/100 mL		
Treatment Dynamic	ES vs. SR 2 Sample *t*-Test	*p*	ES vs. SR ANOVA *p*	ES vs. SR 2 Sample *t*-Test	*p*	ES vs. SR ANOVA *p*	ES vs. SR 2 Sample *t*-Test	*p*	ES vs. SR ANOVA *p*
Pre-Treat	+0.805 SR	<0.001	<0.001	+1.305 ES	0.004	0.003	+13.2 SR	0.602	0.595
Post-Treat	+0.68 ES	0.633	0.602	+1.170 ES	0.010	0.009	+13.8 SR	0.757	0.744
6 Months	+0.823 SR	<0.001	<0.001	+1.212 ES	0.005	0.005	+1.6 ES	0.971	0.970
12 Months	+0.838 SR	<0.001	<0.001	+0.485 ES	0.490	0.488	+27.7 ES	0.369	0.377
18 Months	+0.593 SR	0.033	0.027	+0.28 ES	0.793	0.790	+108.1 ES	0.020	0.022
24 Months	+0.503 SR	0.160	0.136	+0.14 ES	0.909	0.904	+118.3 ES	0.030	0.017

Abbreviations used: μg/L—micrograms/liter, μmol/100 mL—micromoles/100 milliliters, ES—enucleation surgery, SR—stereotactic radiotherapy, ANOVA—analysis of variance.

**Table 3 diagnostics-13-01907-t003:** Statistical study of the inference of patient gender on serum levels of antioxidants, lipid peroxides and albumin thiols.

Males vs. Females	Antioxidants μg/L			Lipid Peroxides μmol/100 mL			Albumin Thiols μmol/100 mL		
Treatment Dynamic	M vs. F 2 Sample *t*-Test	*p*	M vs. F ANOVA *p*	M vs. F 2 Sample *t*-Test	*p*	M vs. F ANOVA *p*	M vs. F 2 Sample *t*-Test	*p*	M vs. F ANOVA *p*
Pre-Treat	+0.045 M	0.821	0.827	+0.995 M	0.029	0.029	+17.7 F	0.479	0.479
Post-Treat	+1.19 M	0.307	0.364	+0.695 M	0.148	0.134	+20 F	0.620	0.637
6 Months	+0.095 F	0.611	0.619	+0.860 M	0.062	0.054	+7.3 M	0.850	0.861
12 Months	+0.037 M	0.885	0.884	+0.716 M	0.302	0.300	+8.6 M	0.785	0.784
18 Months	+0.007 M	0.981	0.983	+2.015 M	0.035	0.031	+71.2 M	0.223	0.168
24 Months	+0.342 F	0.601	0.386	+0.528 M	0.586	0.671	+46.0 M	0.645	0.528

Abbreviations used: μg/L—micrograms/liter, μmol/100 mL—micromoles/100 milliliters, ANOVA—analysis of variance, M—males, F—females.

**Table 4 diagnostics-13-01907-t004:** Comparison of pre-treatment to post-treatment results at study follow-up intervals.

	Antioxidants μg/L					Lipid Peroxides μmol/100 mL					Albumin Thiols μmol/100 mL				
Pre-Treatment Versus	Paired S *t*-Test M. Diff	*p*	Significance	ANOVA *p*	Variance Diff	Paired S *t*-Test M. Diff	*p*	Significance	ANOVA *p*	Variance Diff	Paired S *t*-Test M. Diff	*p*	Significance	ANOVA *p*	Variance Diff
Post-Treat	−0.766	0.238	No	0.242	No	−0.1505	0.120	No	0.644	No	0.3	0.982	No	0.990	No
6 Months	−0.0625	0.239	No	0.650	No	−0.2703	0.001	Yes	0.400	No	−23.4	0.180	No	0.326	No
12 Months	−0.0123	0.836	No	0.987	No	−0.318	0.029	Yes	0.527	No	−23.7	0.093	No	0.682	No
18 Months	0.0958	0.205	No	0.883	No	−0.045	0.805	No	0.484	No	−27.3	0.217	No	0.969	No
24 Months	0.243	0.086	No	0.506	No	0.182	0.307	No	0.067	No	24.7	0.057	No	0.517	No

Pre-treatment serum determinations were statistically compared to post-treatment results using the following tests: paired samples *t* test, one-way ANOVA. The “Significance” column was completed “Yes” if the *p* value presented statistical significance (*p* < 0.05). The *p* values used are presented in the “*p*” column and reflect paired samples *t* tests. The Variance Difference (“Variance Diff”) column was completed yes if a statistically significant variance was found (*p* < 0.05). The corresponding ANOVA *p* values are presented in the “ANOVA *p*” column. Abbreviations used: μg/L—micrograms/liter, μmol/100 mL—micromoles/100 milliliters, Paired S *t*-Test M. Diff—paired samples *t* test mean difference, ANOVA—analysis of variance, Variance Diff—variance difference.

## Data Availability

All data regarding the study population pertains to the Oncological Institute Prof. Dr. Alexandru Trestioreanu Bucharest and is available upon request.

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
