# Peer review of "The Effects of Oncological Treatment on Redox Balance in Patients with Uveal Melanoma"

_diagnostics, 2023, doi:10.3390/diagnostics13111907_

Round 1

Reviewer 1 Report (Previous Reviewer 2)

The authors submitted a research article in which they reported the results of the prospective study for patients undergoing enucleation surgery or stereotactic radiotherapy for uveal melanoma by following systemic oxidative-stress redox markers serum lipid peroxides, total albumin groups and total antioxidant levels. The authors investigated serum levels of antioxidants and lipid peroxides at pre-treatment period and in longer-term follow-up and found that both parameters were elevated. Although the findings seem to be impressive, I would like to make several comments to discuss.

1. The structure of the paper. I found that the structure of the article appears to be a liitle bit of chaotic and difficlt to read. Please, shorten section Introduction, pool all appropriate data according to their origin in the subsections removing description of the methods from the section Results and Discussion.

2. Statistics. Please, arrange this section so that all methods the authors had used would be collected in this section

3. Section Results. Basic characteristics of the patient population should be thoroughly described.

4. Section Discussion. It remained unclear what novelty is and how the authors consider implementing the findings in clinical routine. Please, check and update

5. Section Conclusion. This section requires modification so that it reflect novelity and future perspectives.

Author Response

Thank you for valuable feedback. In response we have revised the manuscript and hope to address the specified feedback question. Please see the attachment.

Reviewer 2 Report (Previous Reviewer 1)

Having read the manuscript, I have the following comments:

1.  The introduction is too verbose and needs to be shortened.

2.  The materials and methods are also too verbose and need to be shortened, you only make mention of what you did if it is different to the procedure has been published.

3.  Tables need revision, they are difficult to read and follow.  I suggest they be in landscape format.  Remove unnecessary significant numbers.

4.  Fig 3 and Table 2 both show the same data, delete that data in Table 2.

5.  Is Fig 4 necessary, because it is unclear what information is supposed to be conveying.

6. L499, it is inappropriate to use the word sex, replace it with the word gender.

7.  Express units as µM or µg/L but not in regards 100 mL.

8.  The discussion is not clear, and why are hypotheses highlighted?

9.  There is some interesting data here why are the antioxidant highest immediately after treatment, however for the other analyses radiotherapy has no effect except for serum albumin thiols, these are items that should be discussed further.  

Some editing of English is required, so please ask someone for whom English is their primary language to assist in editing the manuscript.

Author Response

Thank you for valuable feedback. In response we have revised the manuscript and hope to address the specified feedback questions. Please see the attachment.

Round 2

Reviewer 1 Report (Previous Reviewer 2)

The authors submitted a revised version of the paper along with a clear explanation of the ways by which the corrections were made. I have no serious concerns about the paper in its revised version.

Author Response

On behalf of all authors I extend our sincere gratitude and thanks for your valuable feedback and review process. Thank you.

Paul Filip Curcă,

Reviewer 2 Report (Previous Reviewer 1)

Having read the revised manuscript I have the following comments:

1.  See below for comments regards the English used in the manuscript.

2. Both mL and ml are used throughout the manuscript, please use one of these unit abbreviations.

3. L624 why is there a question mark after Significant and value?

4. L1485 the abbreviation for NADPH oxidase is "NOX" not "nox", please correct.

5. L1493 replace the phrase "sex differences" with "gender differences"

There are example of poorly phrased sentences in the manuscript.  Having a native English speaker will see these sentences being corrected, similarly the tense used varies in the manuscript in sections.  This also needs to be addressed.

Note that when citing multiple authors it is Smith et al., not Smith et all as seen on L91 for example.  Please correct these instances in the manuscript.

L72 what do you mean uveal malign melanoma, don't you mean uveal malignant melanoma?

L825 what do you mean by this heading "dynamics comparison of enucleation surgery (ES) ...."

Author Response

On behalf of all authors thank you sincerely for your valuable feedback. In response we have revised the manuscript and hope to address the specified feedback:

1.  See below for comments regards the English used in the manuscript. - Thank you for highlighting this; we have performed additional grammar and phrase manuscript editing and hope to address this issue. Corrections have been made for et al., uveal malignant melanoma and L825 to "Comparing serum determinations of enucleation surgery (ES) versus stereotactic radiotherapy (SR) patients".

2. Both mL and ml are used throughout the manuscript, please use one of these unit abbreviations. - This has been corrected to mL, thank you. This correction required recreating the graphs for Figures 2 and 3; the Figures have been replaced with corrected versions with mL.

3. L624 why is there a question mark after Significant and value? - The "Significant?" column was completed Yes if p values were <0.05. We have edited the table 1 and 4 legends to better reflect this. We have changed "Significant?" with "Significance".

4. L1485 the abbreviation for NADPH oxidase is "NOX" not "nox", please correct. - This has been corrected; thank you.

5. L1493 replace the phrase "sex differences" with "gender differences" - We have corrected the specified phrase and performed additional corrections for similar instances in the manuscript.

Thank you for your valuable feedback,

Paul Filip Curcă

This manuscript is a resubmission of an earlier submission. The following is a list of the peer review reports and author responses from that submission.

Round 1

Reviewer 1 Report

Having read this manuscript I have the following comments:

1.  What was the rationale for this study, it is not clear from reading the introduction.

2. Why uveal melanoma is a concern to patients wrt vision impairment, no information is given about OS figures for these patients at 5yr post-diagnosis.  This needs to be added.

3. What evidence has been reported that redox balance is affected by uveal melanoma?  The link between the two is not clear.

4.  The description of the various antioxidant assays (P3-4) are verbose and need to be shortened. If you are using a published assay that has not be modified just say X was measured according to that of Smith et al etc.

5.  Either abbreviate litre with a capital "L" or lower case "l" but do not use both throughout this manuscript eg. mL, ml, etc.

6.  Most of the graphs and tables are too small to be read, and when showing statistical values 3 significant numbers is sufficient  unlike that seen in table 9 for example.

7.  Most of the results section repeats the numbers in the Tables and Figures, and if there is no significant difference between treatments then you do not need to state anything more than to say no significant differences were observed between treatments with respect to changes in the concentration of X etc.  By doing this most of the text in the results section can be removed.

8.  How does the information in the 4 graphs in Figs 6-8 differ from each other, are all 4 necessary?  

9.  Fig 1, what does Data mean in the Y axis in the 3 graphs?

10.  L426 you state there is no difference between enucleation and radiotherapy wrt to oxidative stress parameters, which on L428 it states your results are in line with other studies but they are not cited, please add them.  How is this study different to other studies on oxidative stress parameters in uveal melanoma patients?

11.  Please correct Reference 3, 12 14 so they are as per the journal "Diagnotics" guidelines.

Reviewer 2 Report

The authors submitted a research article in wich they reported how they monitored the redox balance via measurement in serum levels of lipid peroxidation markers, total albumin groups and total antioxidant levels via serum iron reduction for patients undergoing uveal melanoma treatment via radiotherapy or enucleation. They prospectively included 39 patients, who underwent treatment for uveal melanoma either by stereotactic radiosurgery or by surgical enucleation. The authors detected serum peroxides and antioxidants statistical significance of pre-treatment and post-treatment values and did not find statistical difference in long-term oxidative stress parameters (up to two years) when comparing uveal malign melanoma treated by surgical enucleation or by stereotactic radiotherapy. Although these findings are empressive, I would like to make some comments to discuss.

1. The authors should give extensive comment for novelty of the study in the section Introduction.

2. The sample size of the study is small, so the data recaived by authors might be misspelled. Please, report this study limitations in connection with prospective study design.

3. It remained unclear whether the dynamics of the antioxidatns / prooxidants profile can reflect a strict similarity in clinical efficacy of both procedures. This fact requires more explanation in the section "Discussion".

Reviewer 3 Report

Dear authors,

This research article is generally well-written, but it needs some editings. 

Editing suggestions;

1) Abstract section should be more detailed.

2) English grammar editing required.

3) It is recommended that the Introduction, material and methods, results, discussion sections should be in bold in the form of a heading at the beginning of the paragraphs.

4) In the Results section; do not repeat in the text information that is already present in the tables.

5) In the discussion part, comments should not be given without discussing the results with the previous study data in the literature.

Best wishes...